# Dearomatization of aromatic asmic isocyanides to complex cyclohexadienes

Bilal Altundas[1], Embarek Alwedi [2], Zhihui Song[3], Achyut Ranjan Gogoi [4], Ryan Dykstra[3], Osvaldo Gutierrez[4] ✉ & Fraser F. Fleming [1] ✉

A dearomatization-dislocation-coupling cascade rapidly transforms aromatic isocyanides into highly functionalized cyclohexadienes. The facile cascade installs an exceptional degree of molecular complexity: three carbon-carbon bonds, two quaternary stereocenters, and three orthogonal functionalities, a cyclohexadiene, a nitrile, and an isocyanide. The tolerance of arylisocyanides makes the method among the mildest dearomatizations ever reported, typically occurring within minutes at −78 °C. Experimental and computational analyses implicate an electron transfer-initiated mechanism involving an unprecedented isocyanide rearrangement followed by radical-radical anion coupling. The dearomatization is fast, proceeds via a complex cascade mechanism supported by experimental and computational insight, and provides complex, synthetically valuable cyclohexadienes.

The dearomatization of substituted arenes is a powerful, complexity-increasing route to diverse carbocycles and heterocycles[1]. Dearomatizations are strategic transformations for assembling complex cyclic, bridged, or spirocyclic scaffolds because multiple bonds are forged from readily available aromatics (**1**)[2–4]. Numerous dearomatizations have featured as key steps in the synthesis of prized bioactive targets[5–9] where brevity is paramount, such as in the syntheses of the anti-viral drug oseltamivir (Tamiflu, **2**)[10] and the pain management opiate oxycodone (**3**)[11] (Fig. 1).

The inherent efficiency of dearomatization strategies, the availability of diverse aromatic precursors, and the rapid increase in molecular complexity, have stimulated several creative arene reduction methods[12]. Historically, the advances in dearomatization have followed organic chemistry's progression from high-energy, two-electron processes (Fig. 2) such as dissolving metal reductions (**1** → **4**)[13], carbene (**5** → **6**)[14] and organometallic[15,16] additions (**7** → **8**), and hydrogenation[17], to more controlled one-electron transfers such as enzymatic (**1** → **9**)[18], oxidative (**10** → **11**)[19], and photochemical (**1** → **12**)[20] reactions.

This manuscript describes a conceptually unexplored dearomatization strategy employing an organometallic addition to promote a single electron transfer (SET) from an electron-rich aromatic sulfur species to initiate the dearomatization process (Fig. 2, part c). An SET from an organometallic-activated arylsulfide generates three components from **13**: the isocyanide-stabilized radical **14**, an alkylarylsulfide (ArSR), and the radical anion **15**. Dislocation of the isocyanide within **15** via a formal [3 + 2]-cycloreversion leads to radical anion **16** whose coupling with **14** affords the functionalized cyclohexadiene **17**. The method is possibly the fastest and mildest dearomatization reported, installs an exceptional degree of molecular complexity, and proceeds through an unprecedented dearomatization–dislocation–radical coupling mechanism.

## Results

### Optimization of the dearomatization process
Experimental forays to develop the dearomatization method were initiated with the cyclopentylisocyanide **19a**, prepared in a single step from the isocyanide building block Asmic (**18**, Fig. 3)[21]. The addition of BuLi to sulfur-substituted isocyanides such as **19a** generates electron-rich sulfuranides **20a**[22] which, in the presence of TMEDA, intercept electrophiles at the isocyanide-bearing carbon to afford trisubstituted isocyanides such as **21**[21]. Speculating that lithium salts might promote scission of the weak isocyanocarbon–sulfur bond, analogous to halogen–lithium exchange reactions[23], led to the inclusion of lithium chloride which completely suppressed the formation of **21a** (**21**,

[1]Department of Chemistry, Drexel University, 3401 Chestnut St., Philadelphia, PA 19104, USA. [2]Merck Inc., 90 E. Scott Ave, Rahway, NJ 07065, USA. [3]Department of Chemistry and Biochemistry, University of Maryland, 8051 Reagents Drive, College Park, MD 20742, USA. [4]Department of Chemistry, Texas A&M University, Ross @ Spence St, College Station, TX 77843, USA. ✉e-mail: og.labs@tamu.edu; fleming@drexel.edu

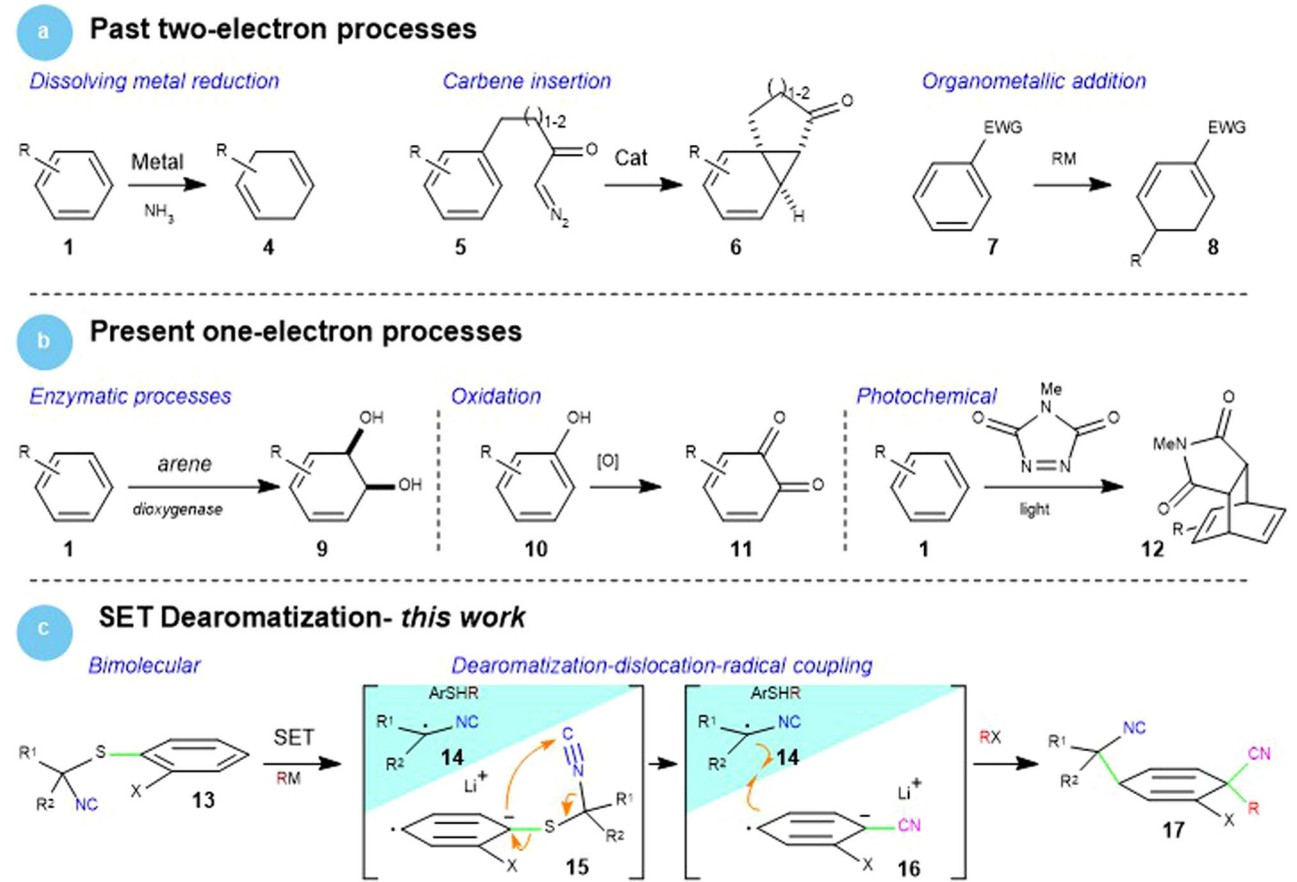

**Fig. 1 | Representative dearomatization-based syntheses.** Dearomatizations can rapidly assemble substituted, six-membered rings for the synthesis of bioactive targets such as Tamiflu (**2**) and Oxycodone (**3**).

**Fig. 2 | Dearomatization methods: past and present. a** Classical dearomatization strategies employing two-electron reductions. **b** Current dearomatization processes employing one-electron processes. **c** This SET approach to dearomatization is followed by NC dislocation and radical–radical anion coupling.

R = Bn) to provide **22a** in 42% yield (Fig. 3); LiBr, LiI, LiOAc, and LiCN were significantly inferior to the use of LiCl. The remarkable formation of **22a** reconstitutes the isocyanide **19a** both in the aromatic core and the cyclopentylisocyanide. The structural reorganization was confirmed by performing an analogous dearomatization–benzylation of **19a** with *p*-bromobenzyl bromide, initially with LiCl and TMEDA but subsequently in 78% yield with HMTETA in the absence of LiCl (see below), to afford **22b** whose structure was unequivocally determined by crystallographic analysis (Fig. 3).

Further optimization led to a reappraisal of the role of lithium salts by focusing on complexing agents to change the nature of the sulfuranide **20a**. While HMPA and the crown ether Kryptofix 222 proved deleterious, the addition of the TMEDA homolog, PMDTA (pentamethyldiethylenetriamine)[24] afforded **22a** in 59% yield. Slow addition of BuLi over 10 min, rather than 30 s, to a solution of **19a** and PMDTA provided a small gain to afford **22a** in 62% yield. Employing the next highest TMEDA homolog, HMTETA

(hexamethyltriethylenetetramine)[25], afforded **22a** with minimal dependence on the addition rate; only one diastereomer was observed in the crude reaction mixture (vide infra).

**Exploring the scope of the dearomatization**
The HMTETA-promoted dearomatization protocol provided a rapid route to a range of complex cyclohexadienes (Fig. 4). Varying Asmic derivatives were subjected to the dearomatization cascade and trapped with different electrophiles to explore the synthetic potential and substrate scope. Specifically, the use of **19a** with allyl bromide or propyl iodide afforded isocyanides **22c** and **22d**, respectively, with the stereoselective installation of the nitrile-bearing quaternary center as a single diastereomer.

Terminating the dearomatization through protonation or deuteration afforded the 1,3-cyclohexadienes **23a** and **23b**, consistent with electrophilic trapping of a cyclohexadienylnitrile anion on the γ-carbon[26]. Analogous addition–dearomatization–benzylation

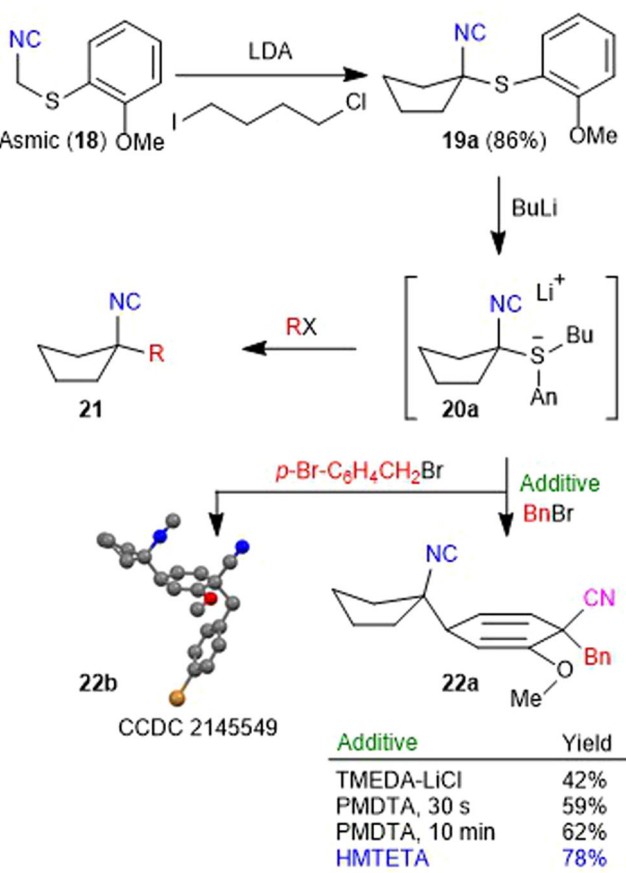

**Fig. 3 | Optimization of the dearomatization for isocyanide 19a.** Alkylation of Asmic (**18**) provides the core scaffold that was dearomatized via sulfuranide **20a** to cyclohexadienes **22a** and **22b**.

sequences were equally efficacious with the 6- and 7-membered Asmic derivatives affording **22e** and **22f**, respectively. Acyclic Asmic derivatives with methyl, propyl, and hexyl substituents similarly afforded, after benzylation, the corresponding 1,4-cyclohexadienes **22g**–**22i**; protonation afforded the 1,3-cyclohexadiene **23c**.

Diversification in the aromatic ring was explored with Asmic analogs bearing 2-fluorophenyl, 5-chloro-2-methoxyphenyl, and 2, 6-dimethoxyphenylsulfanyl substituents and with two isocyanonaphthylene analogs (Fig. 5); an *ortho* heteroatom appears to be required for the dearomatization as attempts to perform the dearomatization with the *p*-methoxy analog of **19a** cleanly returned unreacted material[27]. A brief optimization with the cyclopentyl- and dipropylfluorophenylisocyanides **24a** and **24e** identified the TMEDA–LiCl combination as more efficacious than HMTETA; in comparable dearomatizations with **24e**, HMTETA afforded **25e** in 56% compared to 84% with TMEDA–LiCl. Under optimized conditions, the dearomatization-benzylation afforded the 1,4-cyclohexadiene **25a** as a single diastereomer in 82% yield while the analogous allylation afforded **25b**. The only significant difference in the reaction of the fluorophenylisocyanides **24** compared with Asmic derivatives **19** (Fig. 4) was the need for an extended time for the electrophilic trapping; the fluorine-substituted cyclohexadienyl anion was significantly less nucleophilic than that generated from **19**.

Analogous addition–dearomatization–benzylations with the cyclic 6- and 7-membered fluorophenylisocyanides (**24b** and **24c**) afforded the corresponding cyclohexadienes **25c** and **26d**, respectively; representative acyclic fluorophenylisocyanides (**24e**–**24g**) afforded **25e**–**25g**. Extending the sequence to a 5-chloro-2-methoxy-substituted isocyanide (**24h**) afforded the vinylchloride–vinylether **25h** while the dimethoxyphenyl-substituted isocyanide (**24i**) generated the bis-

vinylether **25i** in which the congested quaternary center is flanked by two '*ortho*' substituents. Dearomatization of isocyanonaphthalenes proceeded with the same reactivity profile as the phenyl analogs affording **25j** for the dearomatization–benzylation, and **25k** for the dearomatization-protonation.

## Computational analysis of the dearomatization mechanism

The dearomatization–dislocation–radical coupling sequence is remarkable in installing three different functionalities and two quaternary centers in a single synthetic operation. Early mechanism-driven optimization seemed most consistent with a SET radical process, particularly the sensitivity toward concentration, regioselectivity preferences, the preference for THF as the solvent, and subsequent mechanistic experiments (vide infra). Valuable mechanistic insight was obtained from dispersion-corrected DFT calculations (for representative examples see ref. 28) noted as UB3LYP-D3/def2-TZVPP-SMD(THF)//UB3LYP-D3/def2-SVP-SMD(THF) that provided structural information on the role of HMTETA complexes and putative intermediates (Fig. 6; the Cartesian coordinates are available in the Supplementary Data file).

Prior calculations on the sulfur–lithium exchange of dialkylAsmic provided the starting point for computationally probing the influence of chelation[22]. Specifically, the importance of HMTETA to sequester the lithium cation was incorporated into the energetics for the complexation and transfer of the alkyl group from MeLi (as a computational surrogate for BuLi with fewer rotatable bonds) to sulfur (Fig. 6, **A** → **A'**); experimentally, substituting MeLi for BuLi leads to only recovered material without any trace of dearomatization which may be due to complex aggregate equilibria in solution[29]. As shown in Fig. 6, an initial HMTETA–Li complexation facilitates the methyl transfer from MeLi to sulfur in the dimethylAsmic complex **A'** leading to the formally anionic sulfuranide **A0** (in complexation with Li⁺•••HMTEA) via a low barrier transition state **TS0** (barrier ~4.4 kcal/mol from complex **A'**). Lithium complexation appears to prevent heterolytic cleavage of **A0** to a solvent-separated isocyanide anion, analogous to related lithium complexing agents in similar equilibria[30]. From sulfuranide **A0**, concurrent decomplexation and homolytic cleavage of the S-alkyl bond generates the anisole radical anion **B•⁻** (akin to those proposed in dissolving metal reductions) and the isocyanide-stabilized radical (**R'**)[31]. Subsequent single electron transfer (SET) from radical anion **B•⁻** to dimethylAsmic **A**[32], triggers the formation of an alkylarylsulfide **B** and radical anion **A•⁻** calculated to be 2.4 kcal/mol downhill with a SET barrier of 1.0 kcal/mol (see SI)[33]. Experimentally, the isolation of AnSBu in virtually quantitative yield is consistent with the computed conversion of **B•⁻** to alkylarylsulfide **B**. From the key thioaryl radical anion **A•⁻**, a facile 5-*endo*-dig cyclization (only a 6.4 kcal/mol barrier) was found via **TS1** to the spirocyclic radical anion **C**. Subsequent fragmentation of the thiazoline via **TS2** completes an unusual isocyanide-to-nitrile dislocation with the release of thioacetone (**D'**) and formation of radical anion **D**[34]. Consistent with these computations, was the isolation of benzylcyclopentenylsulfide from the putative deprotonation of cyclopentathione followed by trapping with benzyl bromide (see **31**, Fig. 7b).

The nitrile-stabilized radical anion **D** is a persistent anionic radical that has featured in several radical–radical anion couplings[35,36]. Radical coupling of **D** with the isocyanide-stabilized radical **R'** (from the homolytic cleavage of **A0**) affords the highly delocalized dienylnitrile anion **E** via **TS_{D–E}**, similar to delocalized anions formed by dissolving metal reduction[26]. The spin density calculations shown in Fig. S3 indicate a higher spin density at the carbon para to nitrile as compared to other positions, which is consistent with the observed coupling regioselectivity[37]. The final bond-forming event, electrophilic trapping of **E**, is highly diastereoselective, congruent with steric shielding by the isocyanide which directs electrophilic attack to the opposite face via **TS3**; experimentally, the nitrile and isocyanoalkyl substituents are located *cis* in the cyclohexadiene.

**Fig. 4 | Scope of the dialkylAsmic dearomatization.** All yields are of pure, isolated materials following chromatographic purification.

## Experiments supporting the proposed mechanism

Two experiments support the computationally derived mechanism (Fig. 7). Performing the dearomatization of **19c** and **24f** in the presence of the radical trap diphenylethylene in each case afforded the pyrroline **27**, consistent with forming an isocyanide-stabilized radical (Fig. 7a). Radical capture of diphenylethylene by the isocyanide-stabilized radical **14a** to afford **26** followed by cyclization and hydrogen atom abstraction leads to **27**; no dearomatization was observed. A separate experiment strongly supports the mechanistically reasonable, yet previously unknown, 2,3-type cycloreversion of **C** via **TS2** to **D** (Fig. 6). Sequential exposure of the dihydro-1,3-thiazene **28** to BuLi and BnBr afforded nitrile **30** and vinyl sulfide **31** (Fig. 7b), paralleling the fragmentation of **C** to **D**; deprotonation of **28** to generate **29** likely triggered cleavage to a metalated nitrile and cyclopentathione whose benzylations led to the benzyl nitrile **30** and the benzyl vinyl sulfide **31**, respectively.

The mechanistic insight was invaluable for developing a crossed dearomatization with two different isocyanides, **19c** as the SET donor and **24i** as the electron acceptor (Fig. 8). Addition of BuLi to a mixture of **19c** and **24i** afforded the cyclohexadiene **25l** consistent with preferential complexation of BuLi to the more Lewis basic methoxy group in **19c** followed by selective SET to the more electron deficient aromatic **24i**. Subsequent fragmentation to the radical anion **16i** and coupling with the isocyanide radical **14a**, generated from **19c**, selectively afforded **25l** (Fig. 8). The crossed dearomatization of **19c** with **24i** demonstrates the viability of expanding the dearomatization from a process requiring two equivalents of one isocyanide to a general crossed union of two different components.

The dearomatization–dislocation-coupling of Asmic isocyanides is unprecedented. The dearomatization is among, if not the fastest

dearomatization method known and yet is differentiated from the current suite of dearomatization reactions by tolerating the rather delicate isocyanide functionality. The dearomatizations with isocyanide-containing arenes overcome the difficulty in disrupting the stable aromatic core while maintaining the fragile electronic framework of isocyanides, the precise attribute underlying their exceptional reactivity in multi-component[38], insertion[39], addition, and radical reactions[40,41].

A series of experimental and computational probes strongly support a unique mechanistic paradigm with the synthetic potential to open rapid routes to highly functionalized scaffolds. The dearomatization–dislocation–radical coupling unifies two arylisocyanide components to rapidly assemble a high degree of molecular complexity that forges three carbon–carbon bonds, two quaternary stereocenters, and nitrile and isocyanide functionality in a single operation. The results lay the foundation for exploiting the cascade in expeditiously building complex cyclohexadienes that are inaccessible via current dearomatization methodologies and establishing mechanistic pathways for advancing unexplored bond-forming strategies.

## Methods
### Experimental procedures

All nonaqueous reactions were performed in flame-dried glassware under a nitrogen atmosphere using Schlenk line techniques. All chemicals were purchased from commercial vendors and used as received unless otherwise specified. Anhydrous tetrahydrofuran (THF) was distilled from benzophenone–sodium under nitrogen before use. Anhydrous HMTETA was prepared by vigorously stirring with $CaH_2$ under nitrogen for 10 min, allowing the $CaH_2$ to settle, and then withdrawing the HMTETA from the

**Fig. 5 | Dearomatization of aryl-substituted dialkylAsmics 24.** All yields are of pure, isolated materials following chromatographic purification (ª 1.1 Equiv of HMTETA was used in place of TMEDA and LiCl).

top of the solution. Anhydrous TMEDA was prepared by vigorous stirring with $CaH_2$ under nitrogen for 10 min, allowing the $CaH_2$ to settle, and then withdrawing the TMEDA from the top of the solution. Stock solutions of anhydrous LiCl in anhydrous THF were prepared as follows: LiCl (500 mg) was flame dried under vacuum (heating three times for 2 min each time), allowed to cool to rt, and then sparged with nitrogen (×3). Anhydrous THF (15 mL) was added and then the solution was stirred until the LiCl was completely dissolved to generate a 0.79 M solution.

Reactions requiring heating were heated in an oil bath. Reactions were magnetically stirred and monitored by thin layer chromatography (TLC) with glass-backed, 250 μm thickness, F254 hard layer SiliaPlate TLC Plates purchased from Silicycle. TLC plates were visualized using a UV lamp (254 nm) and staining with *p*-anisaldehyde or potassium permanganate solutions. Flash chromatography was performed with the Buchi Reveleris X2 Automated Flash Chromatography System using SiliaSep™ Flash 40–63 μm, 60 Å silica gel cartridges or manually with SiliaFlash® P60 40–63 μm, 60 Å irregular silica gel purchased from Silicycle.

## Data analysis

¹H NMR and ¹³C{¹H} NMR high-resolution nuclear magnetic resonance spectra were recorded on a Varian Mercury Plus 400

(400 MHz/101 MHz) or a Varian Unity Inova 500 (500 MHz/ 126 MHz) spectrometer at room temperature. Chemical shifts are reported relative to TMS ($\delta$ 0.00) or $CDCl_3$ ($\delta$ 7.26) for ¹H NMR and $CDCl_3$ ($\delta$ 77.2) for ¹³C{¹H} NMR. IR spectra were recorded as thin films (PerkinElmer Spectrum 100 FT-IR Spectrometer). High-resolution mass spectra were obtained on a Thermo-Electron LTQ-FT 7T Fourier transform ion cyclotron resonance (FT-ICR) spectrometer with an atmospheric pressure chemical ionization (APCI) source with direct infusion run in positive ion mode at 5 kV. Additional accurate mass measurement analyses were conducted on either a Waters GCT Premier, time-of-flight, GCMS with electron ionization (EI), or an LCT Premier XE, time-of-flight, LCMS with electrospray ionization (ESI). Samples were taken up in a suitable solvent for analysis. The signals were mass measured against an internal lock mass reference of perfluorotributylamine (PFTBA) for EI-GCMS, and leucine enkephalin for ESI–LCMS. Waters software calibrates the instruments, and reports measurements, by use of neutral atomic masses. The mass of the electron is not included.

## Detailed experimental procedures

Detailed experimental procedures are provided in the Supplementary Information.

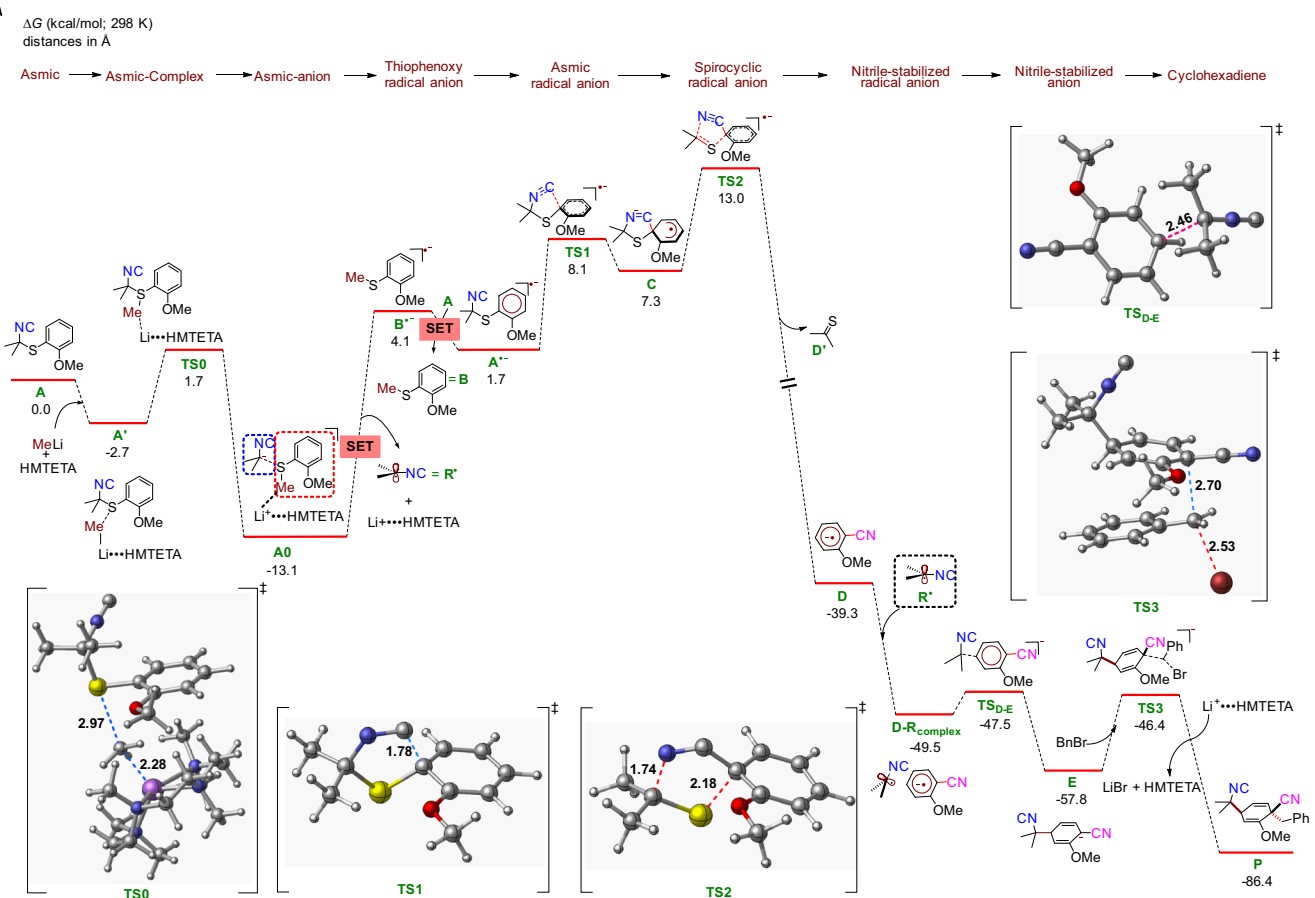

**Fig. 6 | Calculated energetics of dearomatization–dislocation-coupling sequence.** Free energies (kcal/mol) were computed at the UB3LYP-D3/def2-TZVPP-SMD(THF)// UB3LYP-D3/def2-SVP-SMD(THF) level of theory.

**Fig. 7 | DialkylAsmic dearomatization mechanism. a** Trapping with diphenylethylene is consistent with the formation of radical intermediates. **b** The formal [3 + 2] cycloreversion of **28** is consistent with the computationally determined dislocation sequence.

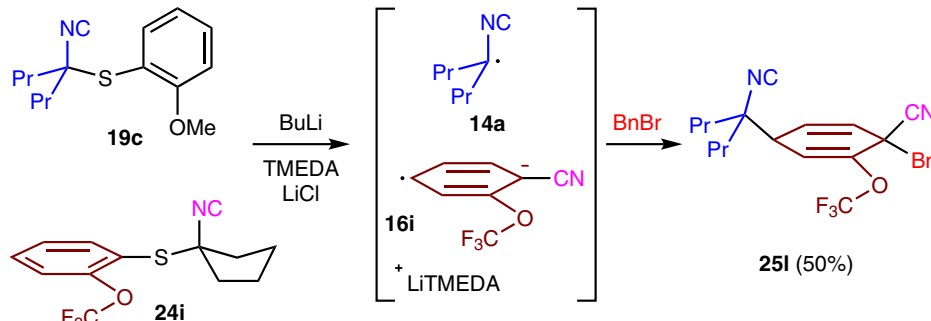

**Fig. 8 | Crossed dearomatization with a matched isocyanide pair.** Electron-rich **19c** serves as the isocyanide donor and the electron-deficient **24i** as the aromatic acceptor in forming cyclohexadiene **25l**.

## Computational analyses

Cartesian coordinates of the optimized geometries and vibrational frequencies of the optimized structures are provided in a separate file in the Supplementary Information.

## Data availability

All data generated or analyzed during this study are provided in the supplementary information files. The X-ray crystallographic coordinates for **22b** reported in this study have been deposited at the Cambridge Crystallographic Data Center (CCDC, #2145549). The data can be obtained free of charge from The Cambridge Crystallographic Data Centre via www.ccdc.cam.ac.uk/data_request/cif.

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

## Acknowledgements

Financial support for the experimental component initially from Drexel University and later from NSF (#1953128, F.F.F.), is gratefully acknowledged. Financial support is gratefully acknowledged from the Texas A&M University (O.G.), the NIGMS of the NIH (R35GM137797, O.G.), Welch Foundation (A-2102-20220331, O.G.), and Camille Dreyfus Teacher Scholar Award (O.G.) for funding, and computational resources from the Texas A&M University HPRC resources (https://hprc.tamu.edu, OG), UMD Deepthought2 (O.G.), MARCC/BlueCrab HPC clusters (O.G.) and XSEDE (CHE160082, O.G. and CHE160053, O.G.). Crystallographic analyses performed by Patrick J. Carroll, HRMS analysis conducted by Timothy P. Wade, Andrew Greene, and Hannah Palmer, and discussions with Dr. Caleb Holyoke are gratefully acknowledged.

## Author contributions

B.A. optimized the dearomatization, developed numerous mechanistic experiments, enlarged the reaction scope, and was involved in manuscript editing. E.A. discovered the dearomatization and performed initial mechanistic experiments. Z.S., A.R.G., and R.D. performed the computations and provided computational figures. O.G. designed the computational analysis and contributed to the manuscript revisions. F.F.F. conceived the experimental design and wrote the manuscript. All authors have given approval for the final version of the manuscript.

## Competing interests

The authors declare no competing interests.
