## [Peer Review File · Nature Communications]

REVIEWER COMMENTS

Reviewer #1 (Remarks to the Author):

The authors report a dearomatization-dislocation-dimerization cascade reaction leading to the diastereoselective synthesis of highly functionalized cyclohexadienes. The method realized the dearomatization with the tolerance of arylisocyanides, the transformation of isocyanides to cyanides, and the rapid construction of molecular complexity. An SET-initiated mechanism involving the isocyanide rearrangement and subsequent radical-radical anion coupling was proposed and evidenced by both convincingly experimental and computational studies. The findings are interesting and representing an elegant extension of the author's previous work. The manuscript is well-organized and the data are high-calibre. Although the substrates and products described in the manuscript may be somewhat specialized, the conceptually new dearomatization strategy will be complementary to cause general interest in chemistry community. After an overall evaluation, the reviewer recommends its publication in Nature Communications after several issues proposed below are settled.

- 1) 20a is the key intermediate for the initiation of the dearomatization cascade reaction. How is the stability of this intermediate? Is it possible to characterize it using $^7\text{Li-NMR}$ or isolate it with the involvement of HMTETA? These attempts deserve to be done and should better be shown in the manuscript.
- 2) All substrates used in the work have an ortho substituent such as OMe and F. What will happen if they are absent? Is there an exact role of this ortho substituent in the reaction?
- 3) MeLi was used as a surrogate for BuLi in the computational study. Is it experimentally available using MeLi?
- 4) The X-ray structure of 22b and the experimental details for XRD analysis should be involved in the supporting information.

Reviewer #2 (Remarks to the Author):

In the manuscript entitled "Dearomatization of Aromatic Aromatic Isocyanides to Complex Cyclohexadienes" by Gutierrez, Fleming and co-workers, they described an intriguing and rare dearomatization of aromatic aromatic isocyanides based on their previous work by slightly modifying the reaction conditions. This work can efficiently obtain a series of polysubstituted cyclohexadiene structures containing nitrile group that are difficult to achieve via other methods. Meanwhile, the

authors investigated this transformation in detail through both experimental studies and DFT calculations. Although the proposed mechanism contains multiple steps with several active intermediates involved, their mechanistic studies support a dearomatization-dislocation-dimerization process. Overall, this transformation is a meaningful addition to the dearomatization of arenes and may attract the attention of researchers in related fields. Therefore, this manuscript should be considered for its accepting in Nat. Commun. after revisions:

1) Although mechanistically distinct, this manuscript has a limited substrate scope. It is recommended to further explore the substrate scope and provide more examples in Fig 4 and 5. More importantly, substituents on the benzene ring would be essential. For instance, have the authors tried naphthalene system?

2) This one agrees that high level of complexity could be built via this process. It would be more useful if the authors could demonstrate some further derivatization on the products.

Reviewer #3 (Remarks to the Author):

The manuscript "Dearomatization of Aromatic Asmic Isocyanides to Complex Cyclohexadienes" by Gutierrez and Fleming groups consists of an experimental investigation regarding the synthesis, followed by a theoretical investigation. Though, this looks like good scientific work overall, but the manuscript is not easy to read. I recommend the author to reorganize the manuscript before publication. As a theoretician, I can mostly judge on the computational part. Here, I see some deficiencies in the computational description that requires some attention. I suggest to resubmit the manuscript after major revision of the following points.

1. The TOC. The radical on the third structure should move close to the ring.
2. I think the "radical radical coupling" will be more acceptable than the "dimerization".
3. Figure 1, the arrows are on the opposite direction. The current one looks like the aromatization process.
4. The reference 36 is not necessary here. I cannot see any reference values for this paper.
5. And I am quite surprised that none of the computational models and used programs are cited in the main paper, and I also did not find them in the Supporting Information. That is not appropriate nowadays, particularly, for a paper which presents important computational results.

6. Also, the final energies require cross-checking. This cross-checking and improvement in the theory level is absolutely unavoidable. The M06-2X functional maybe more appreciate for this pure organic reaction.

7. In Figure 6, the energy of A⁻ is 1.7? or -1.7? It looks like a -1.7 according the scale of the figure. Please double check this energy.

8. There are two kinds of SET processes included in this mechanism. The inner-sphere SET and outer-sphere SET. The inner one is from A⁻ to B⁻, where the radical R generated. The reaction barrier for this radical generation process should be calculated. The transition state from D to E related to the radical radical coupling process is also missing in this manu. The outer-sphere SET is the one from B⁻ to B, and the energy could also be estimated by the Marcus theory. As a reader, I would like to see the reaction barriers of three SET processes.

9. The 3D structure is not clear.

10. The coordinates of the structures from computational chemistry should be provided in SI, together with the details of computational methods.

Reviewer 1

- The anionic sulfuranide intermediate **20a** is only stable below -50 °C which makes isolation and characterization a challenge. Please note that in the original submission **20a** was referred to as a "sulfuranylde" which we have revised throughout the manuscript to "sulfuranide" following recent advice from IUPAC. We have been using ¹³C NMR to explore the structure of sulfuranides analogous to **20a** with limited success. The problem is partly that the signals are weak, particularly the diagnostic isocyanide carbon, and partly the presence of several unidentified solution species; the problems are compounded by limited stability. We are continuing to determine the structure of these isocyanide-stabilized sulfuranides but because of the experimental complexity we plan to submit our findings as a separate manuscript.
 - We have previously prepared a related anionic sulfuranide by adding a sulfide to a lithiated alkylisocyanide as described in reference 25. We have therefore moved reference 25 to the associated text on page 5 so that the interested reader can consult the appropriate research: "Addition of BuLi to sulfur-substituted isocyanides such as **19a** generate electron rich sulfuranides **20a**²⁵..."
- ortho* heteroatom substituents appear to be required for pre-complexation and delivery of the alkyllithium to sulfur. We have added the following text on page 8 to clarify: "Diversification in the aromatic ring was explored with Asmic analogs bearing 2-fluorophenyl, 5-chloro-2-methoxyphenyl, and 2, 6-dimethoxyphenylsulfanyl substituents (Fig. 5); *an ortho heteroatom appears to be required for the dearomatization as attempts to perform the dearomatization with the p-methoxy analog of 19a cleanly returned unreacted material.*³⁵" New reference 35 [*Org. Lett.* **20**, 5910-5913 (2018)] has been added which is a detailed survey of *ortho* substituents required for accessing sulfuranides in the context of alkylations.
- Experimentally, substituting MeLi for BuLi leads to only recovered material without any trace of dearomatization. We have added the text "*Substituting MeLi for BuLi leads to only recovered material without any trace of dearomatization*" as a new reference (#40) on page 11, first line "... transfer of the alkyl group from MeLi (as a computational surrogate for BuLi)⁴⁰..."
- The CIF data for **22b** has been provided as supporting information.

Reviewer 2

1. The suggestion to dearomatize isocyanonaphthalenes led us to prepare and expose two new substrates to the dearomatization conditions. The two new examples, **25j** and **25k** (Figure 5), illustrate that both deconjugation-benzylation and deconjugation-protonation proceed with isocyanonaphthalenes in the same fashion as the corresponding isocyanobenzenes.
2. We agree with reviewer 2 that the cyclohexadienes seem well-suited to further derivatization. We have tried to cyclize the enol ether onto the isocyanide which appears to proceed but with rather complex reaction mixtures. Our plan is to convert the isocyanide into a nitrilium to facilitate the cyclization which we hope to report in a subsequent manuscript expanding on the dearomatization chemistry.

Reviewer 3

1. a. The suggestion is made to reorganize the manuscript. We hope that the collective revisions make the current organization better to understand which reviewer 1 describes as "well-organized."
b. The radical on the third structure in the TOC has been moved closer to the ring.
2. The term "dimerization" has been replaced throughout by either "coupling" or "radical-radical coupling."
3. The arrows in figure 1 are disconnection arrows so are correct as drawn.
4. Reference 36 has been deleted.
5. We sincerely apologize for forgetting to submit the computational details as an SI file. The computational modeling and relevant details have been included as a separate SI file in the revised submission.
6. We agree with the reviewer and have performed calculations with the single point energies with M06-2X functional. We have added the relevant energies into SI for the resubmission. Overall, the conclusions remain the same.
7. We thank the reviewer for pointing out this problem and apologize for the inappropriate scale of the energy in the figure. We have checked our data and revised the energy of A⁻ from -1.7 to 1.7 in Figure 6.
8. We agree and thank the reviewer for the suggestion to evaluate inner- and outer-sphere SET processes. We have calculated the relevant barriers for the three processes. For the first process i.e. inner sphere SET as described by the reviewer, the barrier was calculated to be 5.6 kcal/mol with respect to our reference species A (18.6 kcal/mol)

with respect to the preceding intermediate). The relevant transition state for the radical-radical coupling process has been located with the process proceeding via a low barrier (barrier ~2 kcal/mol). For the last barrier, from B^{·-} to B, the SET barrier was found to be 5.1 kcal/mol with respect to our reference species A (1 kcal/mol with respect to the preceding intermediate). We thank the reviewer for kind suggestion about Marcus theory; we have calculated the relevant SET barriers in accordance with Marcus theory. The relevant energy barriers are now included in the manuscript as well as in the SI. The relevant references for the methods of calculation are also included in the SI.

9. The three-dimensional structures of the transition structures have been redrawn and placed in the previous white space. In combination with the associated ChemDraw images, we believe the images provide a good composite picture.
10. As suggested, we have added the coordinates of the structures and the details of computational methods into the revised SI.

REVIEWERS' COMMENTS

Reviewer #1 (Remarks to the Author):

1) The authors' answer to Q3: We have added the text "... " as a new reference (#40). Please check carefully because it is Ref.38 in the revised manuscript. This means MeLi does not work experimentally, which will vastly reduce the reliability of calculations on the dearomatization mechanism (Figure 6).

2) The X-ray structure of 22b and the experimental details for XRD analysis (crystallographic information including crystal preparation details, data collection and refinement details, etc...and at least Table 1 of Crystal data and structure refinement) should be added "in" the supporting information file. If the authors have no experience, please ask colleagues around for instructions.

Reviewer #2 (Remarks to the Author):

The authors have addressed all the questions/suggestions in the previous version of manuscript. The current one should be acceptable by Nature Common.

Reviewer #3 (Remarks to the Author):

The revised manuscript has addressed all the questions. Now I would like to support its publication in NC.

Reviewer 1

- 1) The authors' answer to Q3: We have added the text "... " as a new reference (#40). Please check carefully because it is Ref.38 in the revised manuscript. This means MeLi does not work experimentally, which will vastly reduce the reliability of calculations on the dearomatization mechanism (Figure 6).

Response.

The reference concerning the substitution of MeLi for BuLi, now number 35, has been changed from: "Substituting MeLi for BuLi leads to only recovered material without any trace of dearomatization" to "MeLi was substituted for BuLi in the calculations to accelerate the computations by decreasing the number of conformations (three rotatable bonds). Experimentally, substituting MeLi for BuLi leads to only recovered material without any trace of dearomatization which may be due to complex aggregate equilibria in solution: Jones, A. C., Sanders, A. W., Bevan, M. J. & Reich, H. J. Reactivity of Individual Organolithium Aggregates: A RINMR Study of n-Butyllithium and 2-Methoxy-6-(methoxymethyl)phenyllithium *J. Am. Chem. Soc.* **129**, 3492–3493 (2007)."

- 2) The X-ray structure of **22b** and the experimental details for XRD analysis (crystallographic information including crystal preparation details, data collection and refinement details, etc...and at least Table 1 of Crystal data and structure refinement) should be added "in" the supporting information file. If the authors have no experience, please ask colleagues around for instructions.

Response.

The CIF file has been provided for the structural analysis of **22b** and may also be obtained from the Cambridge Crystallographic Data Centre.